# Understanding the Functional Expression of
${\mathrm{Na}}^{+}$-Coupled SLC4 Transporters in the Renal and Nervous Systems: A Review

**DOI:** 10.3390/brainsci11101276

**Published:** 2021-09-26

**Authors:** Le Du, Aqeela Zahra, Meng Jia, Qun Wang, Jianping Wu

**Affiliations:** 1School of Chemistry, Chemical Engineering and Life Sciences, Wuhan University of Technology, Wuhan 430070, China; ledu@whut.edu.cn (L.D.); Zahra@whut.edu.cn (A.Z.); 2Beijing Tiantan Hospital, Capital Medical University, Beijing 100070, China; jiameng@bjtth.org (M.J.); wangq@ccmu.edu.cn (Q.W.); 3Advanced Innovation Center for Human Brain Protection, Capital Medical University, Beijing 100070, China; 4National Clinical Research Center for Neurological Disease, Beijing 100070, China; 5Health Science Center, Yangtze University, Jingzhou 434023, China

**Keywords:** ${\mathrm{Na}}^{+}/{\mathrm{HCO}}_{3}^{-}$ cotransporters (NBCs), 4,4′-diisothiocyano-2,2′-stilbenedisulfonic acid (DIDS), proximal renal tubular acidosis (pRTA), migraine

## Abstract

Acid-base homeostasis is crucial for numerous physiological processes. ${\mathrm{Na}}^{+}/{\mathrm{HCO}}_{3}^{-}$ cotransporters (NBCs) belong to the solute carrier 4 (SLC4) family, which regulates intracellular pH as well as ${\mathrm{HCO}}_{3}^{-}$ absorption and secretion. However, knowledge of the structural functions of these proteins remains limited. Electrogenic NBC (NBCe-1) is thought to be the primary factor promoting the precise acid–base equilibrium in distinct cell types for filtration and reabsorption, as well as the function of neurons and glia. NBC dysregulation is strongly linked to several diseases. As such, the need for special drugs that interfere with the transmission function of NBC is becoming increasingly urgent. In this review, we focus on the structural and functional characteristics of NBCe1, and discuss the roles of NBCe1 in the kidney, central nervous system (CNS), and related disorders, we also summarize the research on NBC inhibitors. NBCe1 and the related pathways should be further investigated, so that new medications may be developed to address the related conditions.

## 1. Introduction

Intracellular pH (pHi) and extracellular pH (pHo) affect almost all biological processes, including metabolism, protein synthesis and activity, ion-channel activation, neuronal excitability, and the cardiovascular system. Consequently, disturbance of the acid–base balance in the body can lead to various serious diseases [1,2,3]. Acid–base homeostasis is achieved by the balance between transporters that load the cell with acid and those that extrude acid from the cell. The acid loaders include anion exchangers (AEs) [4], electrogenic ${\mathrm{Na}}^{+}/{\mathrm{HCO}}_{3}^{-}$ cotransporters operating with a ${\mathrm{Na}}^{+}:{\mathrm{HCO}}_{3}^{-}$ stoichiometry of 1:3 [5], and ${\mathrm{HCO}}_{3}^{-}$ channels, which mediate the influx of acid equivalents across the cell membrane, as shown in Figure 1A. The acid extruders include $H^{+}$-pumps [6], ${\mathrm{Na}}^{+}/H^{+}$ exchangers (NHEs) [7], and both electrogenic and electroneutral ${\mathrm{Na}}^{+}/{\mathrm{HCO}}_{3}^{-}$ cotransporters operating with a ${\mathrm{Na}}^{+}:{\mathrm{HCO}}_{3}^{-}$ stoichiometry of 1:1 or 1:2, which mediate the efflux of acid equivalents across cell membranes, as shown in Figure 1B. The coordinated activity of these acid–base transporters regulate pHi in mammalian tissues. This coordinated activity keeps cell pH within a restricted physiological range in non-epithelial cells, which is crucial for cell function and survival. All such transporters are not, however, expressed in the same cell. In general, non-epithelial cells express NHEs (mostly the ubiquitous NHE1), AEs (especially AE3), and NBCs. In epithelial cells, the above-mentioned transporters exhibit distinct membrane domain locations, and some transporters show exclusive expression in the apical membrane or basolateral membranes. For example, AE4 is exclusively expressed in the apical membrane, whereas NBCe1 is predominantly expressed in the basolateral membrane of epithelial cells.

Fluctuations in cellular pH have major functional effects, especially in the nervous system, since numerous ion-channel activities related to neurotransmission are sensitive to pHi/pHo changes [8]. It is important to note that passive diffusion and pHi-regulated transporter movement across plasma membranes influence both pHi and pHo. Under the physiological conditions of the CNS, several major ion transporters–including NHE, ${\mathrm{Na}}^{+}/{\mathrm{Ca}}^{2+}$ exchanger (NCX), ${\mathrm{Na}}^{+}/K^{+}/{\mathrm{Cl}}^{-}$ cotransporter (NKCC), and NBC, especially play important roles in regulating ion homeostasis, cell volume, and cell signal transduction. In acute brain diseases–such as traumatic brain injury (TBI), the above-mentioned transporters are rapidly activated and regulate pHi and pHo; ${\mathrm{Na}}^{+}$, $K^{+}$, and ${\mathrm{Ca}}^{2+}$ homeostasis, synaptic plasticity, and myelin formation [9]. In chronic neurological diseases, such as Parkinson’s disease (PD), multiple sclerosis (MS), and Alzheimer’s disease (AD), ion transporters are involved in glial activation, neuroinflammation, and neuronal damages [10]. However, ion transporters play a complex role in modulating clinical phenotypes [11]. While it is well established that all ion transporters are polymers that operate as part of a regulatory network, how these systems themselves function as regulatory components remains uncertain [12]. In other areas of the nervous system, such as the retina, one related set of disorders is retinal dystrophies [13], which are characterized by different mutations across ocular ion-channel genes, leading directly to a wide range of channelopathies, which seems to be related to alterations in neurotransmission (probably linked to ion-channel impairments).

## 2. The SLC4 Family

In the mid-1970s, Boron and Thomas first discovered that ${\mathrm{CO}}_{2}/{\mathrm{HCO}}_{3}^{-}$ provides the active up-hill extrusion of acids or bases [14]. In mammals, ${\mathrm{HCO}}_{3}^{-}$ transporters are expressed throughout the body, and are essential for many physiological processes, including the transport of carbon dioxide (${\mathrm{CO}}_{2}$) from capillaries to pulmonary capillaries, and the secretion or absorption of acid–base equivalents (such as the secretion of HCl in the stomach, ${\mathrm{NaHCO}}_{3}$ secretion in the pancreas, and reabsorption of ${\mathrm{Na}}^{+}$ in the kidneys), as well as for affecting cell volume and pH in nearly every cell. ${\mathrm{HCO}}_{3}^{-}$, is transported and encoded by the SLC4 and SLC26 gene families, respectively; NBCs belong to the SLC4 family. The classification of SLC4 transporters is shown in Figure 2. Mammalian genomes contain 10 SLC4 genes (SLC4A1–5 and SLC4A7–11) that can be divided into three major clades [5,15,16,17]. The ${\mathrm{Na}}^{+}$-independent, electroneutral ${\mathrm{Cl}}^{-}/{\mathrm{HCO}\;}_{3}^{-}$ exchangers are called anion exchangers (AEs), and include AE1 (SLC4A1), AE2 (SLC4A2), and AE3 (SLC4A3). The ${\mathrm{Na}}^{+}$-dependent ${\mathrm{HCO}}_{3}^{-}$ transporters include electrogenic ${\mathrm{Na}}^{+}/{\mathrm{HCO}}_{3}^{-}$-cotransporter (NBCe1/SLC4A4, NBCe2/SLC4A5), electroneutral ${\mathrm{Na}}^{+}/{\mathrm{HCO}}_{3}^{-}$-cotransporter (NBCn1/SLC4A7, NBCn2/SLC4A10) and a ${\mathrm{Na}}^{+}$-driven ${\mathrm{Cl}}^{-}/{\mathrm{HCO}}_{3}^{-}$ exchanger (NDCBE/SLC4A8). The third clade includes ${\mathrm{Na}}^{+}$-coupled borate transporter (BTR1/SLC4A11) and AE4 (SLC4A9), whose functions are unknown.

## 3. The $Na^{+}$-Dependent $HCO_{3}^{-}$ Transporters

### 3.1. NBCe1(SLC4A4)

Boron and Boulpaep [18] first identified the activity of NBCe1 in the basal lateral membranes of proximal renal tubular epithelial cells in vertebrates using electrophysiological techniques. They also characterized the NBC present in the basolateral membrane as electrogenic, dependent on ${\mathrm{Na}}^{+}$ and ${\mathrm{HCO}}_{3}^{-}$, and sensitive to DIDS. Romero et al. [19] used an expression-cloning strategy to effectively clone NBCe1 from salamander renal tubules. This was the first clone of NBC cDNA, now known as NBCe1. Subsequently, researchers have cloned and characterized four orthologs of NBCe1 (NBCe2, NBCn1, NDCBE, and NBCn2) with verified NBC activity, greatly promoting the development of molecular physiology research related to NBCs [20,21,22,23,24].

NBCs represent a relatively new field of research, and knowledge of the structural functions of these proteins remains limited. During the past two decades, some exciting progress has been made in the study of NBCe1′s structural function. Based on the regional anatomy of the red cell anion exchanger AE1, Romero et al. predicted the topological structure of NBC, suggesting that it includes a relatively large cytoplasmic N-terminal domain, a transmembrane domain (TMD), and a smaller cytoplasmic C-terminal domain [25,26]. Moreover, Romero’s model, TMD includes 14 transmembrane regions (TM1–14); of these, 13 (TM1–12 and TM14) are transmembrane alpha helices, while TM13 is a non-alpha-helix transmembrane structure. Meanwhile, Zhu et al. [27] performed an extensive substituted cysteine-scanning mutagenesis analysis to further research NBCe1, and put forward a new model. In this new model, the transmembrane domain (TMD) of NBCe1 also has 14 transmembrane regions, all of which are alpha-helical structures. In 2018, Huynh et al. [28] determined the structure of the membrane domain dimer of human NBCe1 at a 3.9 Å resolution via cryo-electron microscopy. CryEM reconstruction of human NBCe1 protein consists of two monomers which form an identical, double-headed eagle like structure. The gate domain, core domain, extracellular loop 3 (EL3) domain, and the cytoplasmic region of each monomer resemble an eagle’s body, wing, head and foot. The atomic model of the NBCe1 monomer consists of 14 TMs, 4 amphipathic helices (H1–4), a short (single-round) cytoplasmic helix (H5), and the loops connecting all of these helices. Furthermore, this new paradigm is divided into two domains: the gate domain, and the core domain. The gate domain consists of six TMs (TMs 5–7 and 12–14), the amphiphilic helix H4, and the short cytoplasmic helix H5, while the core domain is composed of eight TMs (TMs1–4 and 8–11) and the amphiphilic helices H1–3. The model of NBCe1 is shown in Figure 3. The predicted structure of NBCe1 comprises 1035 amino acids, among which the N-terminal domain includes a core domain and a variable region. Following isolation of a mammalian NBCe1 homolog with a comparable amino acid sequence (labeled NBCe1-A), its variations NBCe1-B through NBCe1-E were identified [29,30]; moreover, these variations differ only in their N- and C-terminal sequences. The N-terminal domains of NBCe1-A and -B are different, where the first 41 amino acids in NBCe1-A are replaced by 85 alternative amino acids in NBCe1-B. NBCe1-C is the same as NBCe1-B, except that the last 61 amino acids are replaced by 46 alternative amino acids. A common feature shared by three NBCe1 splice variants (-A, -B, and -C) is alternative splicing, which results in varying N- and C-termini, that can affect transporter activity and modulation. For instance, the N termini of NBCe1-B and -C (but not -A) include an auto-inhibitory domain that inhibits transporter function.

In vitro and in vivo studies have proven that, in addition to kidney proximal tubular cells, NBCs are found in numerous cell types, including glial cells [31], neurons [32], eye tissues [33], reproductive tract tissues [29], etc. Notably, the three human NBCe1 variations exhibit distinct patterns of tissue expression, intrinsic activity, and gelation mechanisms. At the same time, NBCe1-A (sometimes known as kNBC1) is primarily found in the basolateral membranes of the renal proximal tubules, where it facilitates the secretion of the bulk of bicarbonate from cells [23]. The renal tubular epithelial cells play a role in absorbing ${\mathrm{HCO}}_{3}^{-}$, and NBCe1-A transports ${\mathrm{Na}}^{+}$ and ${\mathrm{HCO}}_{3}^{-}$ from the epithelial cells to the interstitial space at a ratio of 1:3 [34,35]. In contrast, NBCe1-B (sometimes known as pNBC1) is widely distributed in the body, and is mainly expressed in the basement membranes of ${\mathrm{HCO}}_{3}^{-}$-secreted epithelial cells (such as pancreatic duct epithelial cells, enamel cells, and digestive tract epithelial cells) transporting ${\mathrm{Na}}^{+}$ and ${\mathrm{HCO}}_{3}^{-}$ from interstitial tissues to epithelial cells at a ratio of 1:2 [36,37]. A notable exception is that NBCe1-B can be found in numerous non-epithelial cells, including neurons, astrocytes, cardiomyocytes, etc., and NBCe1-C (sometimes known as hNBC1) which is found almost exclusively in the brain and expressed in the CNS, predominantly in astrocytes [38]. Concerning the activity of these three variants, McAlear et al. [39] compared the ${\mathrm{HCO}}_{3}^{-}$ transport activity of NBCe1-A, B, C via the voltage clamp technique, and by microelectrode measurement of intracellular pH, using *Xenopus* oocytes as the heterologous expression system; their results demonstrated that NBCe1-A is four times as active as NBCe1-C.

### 3.2. NBCe2 (SLC4A5)

Human NBCe2 (sometimes known as NBC4) can be functionally characterized as one of the electrogenic ${\mathrm{Na}}^{+}/{\mathrm{HCO}}_{3}^{-}$ cotransporters. Pushkin et al. [40] originally cloned and characterized NBCe2 from the human heart. NBCe2 has 1137 amino acids, 53% of which are homologous with NBCe1. Northern blot analysis was performed using a blot obtained from OriGene Technologies Inc. to determine the expression of NBCe2 in various human tissues. Findings showed high levels of NBCe2 expression in the liver, testes, and spleen [40], and moderate expression levels in the heart, kidneys, placenta, and stomach [41]. Subsequent studies revealed that NBCe2 is most abundant in the liver, where it is expressed in hepatocytes and intrahepatic cholangiocytes in bile ducts [42]. Interestingly, other members of the NBC family are not expressed in the liver, and even the widely expressed NBCe1 is not abundant in this tissue. Of the seven NBCe2 splices in the Human Protein Database [43,44], NBCe2-C is the only variation known to have mutations encoding electrogenic NBC activity. Compared with NBCe1, NBCe2 has cell-type-specific stoichiometry. For instance, NBCe2 exhibits an apparent 1:2 stoichiometry in *Xenopus* oocytes [43], while an apparent 1:3 stoichiometry is expressed in the renal mPCT cell line [41].

### 3.3. NBCn1 (SLC4A7)

NBCn1 was first cloned by Pushkin et al. [45] from human skeletal muscle and later by Choi et al. [46] from rat aorta. Canonical human NBCn1 contains 1214 amino acids, presenting a unique isoform with 59% amino acid identity NBCe1, while its predicted structure is also similar to that of NBCe1. NBCn1 has an apparent ${\mathrm{Na}}^{+}:{\mathrm{HCO}}_{3}^{-}$ stoichiometry of 1:1, and an associated conductance that is carried by ${\mathrm{Na}}^{+}$ at a rate of ~50% [46]. NBCn1 expression is apparent only in the heart and skeletal muscle in humans; however, in rats, NBCn1 mRNA has the highest expression levels in the spleen and testes, and is moderately expressed in the liver, heart, brain, lungs, and kidneys, but not in the skeletal muscle [47]. Interestingly, almost all NBCs are sensitive to DIDS, but NBCn1 has only a modest sensitivity to DIDS, with a maximal inhibition of 25%, which differentiates it from other ${\mathrm{HCO}}_{3}^{-}$ transporters of the SLC4 family. In addition, NBCn1 has ${\mathrm{Na}}^{+}$-channel-like activity, which is uncoupled by the ${\mathrm{Na}}^{+}/{\mathrm{HCO}}_{3}^{-}$ cotransport activity, while DIDS enhances the NBCn1-associated ${\mathrm{Na}}^{+}$-conductance [46]. Similarly to other NBCs, NBCn1 exists as a variety of variants due to alternative promoter sites and splicing events. NBCn1-knockout mice developed blindness and auditory impairment due to the deterioration of sensory receptors in neurons [48].

### 3.4. NDCBE (SLC4A8)

Romero et al. [49] were the first to clone mammalian cDNAs encoding the ${\mathrm{Na}}^{+}$-driven ${\mathrm{Cl}}^{-}/{\mathrm{HCO}\;}_{3}^{-}$ exchanger NDCBE from *Drosophila*. The canonical human NDCBE has 1093 amino acids with 50% and 70% amino acid sequence homology with NBCe1 NBCn1, respectively [50]. NDCBE is the sole obligate ${\mathrm{Na}}^{+}$-driven ${\mathrm{Cl}}^{-}/{\mathrm{HCO}\;}_{3}^{-}$ exchanger, which is different from the AE ${\mathrm{Cl}}^{-}/{\mathrm{HCO}\;}_{3}^{-}$ exchangers AEs. The former serves as an acid extruder, which moves external ${\mathrm{Na}}^{+}$ and ${\mathrm{HCO}}_{3}^{-}$ into cells in exchange for internal ${\mathrm{Cl}}^{-}$; however, the latter serve as acid loaders, which moves external ${\mathrm{Cl}}^{-}$ into the cells in exchange for internal ${\mathrm{HCO}}_{3}^{-}$. Surprisingly, human NDCBE and similar *Drosophila* transporters lack a common DIDS-interaction motif (KXXK) at the extracellular terminus of TM5, but they are extremely susceptible to DIDS inhibition. While human NDCBE possesses such a motif at the putative extracellular end of TM 3, DIDS-sensitive *Drosophila* NDAE1 lacks such a motif at either of these sites. NDCBE transcripts are abundantly detected in the testes and throughout the central nervous system, with very little expression in the kidneys and ovaries [51]. In the brain, the NDCBE protein primarily governs ${\mathrm{HCO}}_{3}^{-}$-dependent acid extrusion in neurons [52]. Xu et al. [53] confirmed that SLCA8 deletion does not lead to significant acid–base imbalance or electrolyte abnormalities in pathophysiological states.

### 3.5. NBCn2 (SLC4A10)

NBCn2 was initially cloned in the murine insulinoma cell line [54] and originally described as a ${\mathrm{Na}}^{+}$-driven ${\mathrm{Cl}}^{-}/{\mathrm{HCO}\;}_{3}^{-}$ exchange protein, identified as NCBE. In 2008, Parker et al. [55] found that, in humans, the protein encoded by the SLC4A10 gene usually performs electroneutral ${\mathrm{Na}}^{+}/{\mathrm{HCO}}_{3}^{-}$ exchange and ${\mathrm{Cl}}^{-}$ self-exchange functions, with the latter occurring under non-physiological conditions. Parker et al. identified SLC4A10 as an electroneutral ${\mathrm{Na}}^{+}/{\mathrm{HCO}}_{3}^{-}$ transporter, and named the protein NBCn2. In humans, NBCn2 has 1118 amino acids, with 71% amino and 65% acid sequence homology with NDCBE and NBCn1, respectively. Original studies SLC4A10, notably in post-synaptic membranes, have shown prevalent expression in the brain and in the choroid plexus epithelial cells [56]. NBCn2 plays a very significant role in the CNS, regulating the pHi of neurons and participating in cerebrospinal fluid secretion [57]. It has been suggested that SLC4A10 gene deletion is related to human autism [58]. In humans, the translocation of this gene can cause symptoms such as partial epilepsy, intellectual disability, and cognitive impairment [59,60]. SLC4A10-knockout mice display greatly reduced ventricular volume, increased incidence of epilepsy, and reduced sensitivity to predation activities [59].

## 4. Physiology of NBCe1

### 4.1. NBCe1 in Renal Acid–Base Regulation

The kidneys are an extremely important organ for filtration and reabsorption, and their major task is to maintain acid–base homeostasis. The kidneys excrete non-volatile acidic substances in the urine, mainly in the form of $H^{+}\;$and ${\mathrm{NH}}_{4}^{+}$. They restore ${\mathrm{HCO}}_{3}^{-}$ to the bloodstream through a reabsorption process to maintain systemic acid–base homeostasis. The proximal tubule is the most important site of ${\mathrm{HCO}}_{3}^{-}$ reabsorption in a nephron, responsible for recovering ~80% of the ${\mathrm{HCO}}_{3}^{-}$ in the glomerular filtrate through the proximal tubular epithelial cells into the blood, while the remaining 20% is absorbed in the medullary thick ascending limb (mTAL) and collecting tubules.

As mentioned above, NBCs are expressed in both epithelial and non-epithelial cells. The direction of NBC’s movement is one of the significant variations between their mode of function in the kidneys and in other tissues. The NBCs in kidney proximal tubule cells mediate ${\mathrm{HCO}}_{3}^{-}$ secretion into the blood, whereas in other cells and organs—such as the liver and heart—NBC transporters mediate ${\mathrm{HCO}}_{3}^{-}$ secretion from the blood into the cells. Therefore, NBC activity in the proximal tubules of the kidneys causes cell acidification. However, in some other tissues, NBCs can cause cell alkalization. Two possibilities have been raised to explain why the direction in which NBCs move in the kidney differs from that in other tissues: (1) the difference in the direction is due to the fact that kidney cells have relatively different membrane potentials or cell ionic compositions; (2) kidney cells may express a different NBC isoform compared to other tissues [61]. Some studies have provided evidence to support both possibilities. According to molecular studies, NBCs in the proximal tubules are different from cardiac NBCs [23], possessing opposite functional modes of cotransport (that is, the efflux of kidney cells and the influx of heart cells, respectively). This might be due to two distinct isoforms in these tissues. However, the identical NBC isoform is expressed in both tissues, but their functions are exerted in opposite directions. In both proximal tubular kidney cells [62,63] and pancreatic cells [20,36], NBCe1 is expressed, but acts in efflux mode in the proximal kidney tubules, whereas, due to the depolarized membrane potential caused by CFTR activation and subsequent ${\mathrm{Cl}}^{-}$ secretion [64], the same transporter operates in the influx mode in the pancreatic duct cells. Figure 4 shows the different transport modes of different NBCe1 variants in the pancreatic duct and renal proximal tubule cells.

Three variants of NBCe1 exist in the kidneys: NBCe1-A, NBCe1-B, and NBCe1-D [29]. Historically, NBCe1-A was considered the only renal NBCe1 variant. Recently, NBCe1-B was confirmed to be expressed in renal cells. Furthermore, an NBCe1-B-knockout (KO) mouse model was generated by Brady et al. [65] using CRISPR/Cas9 gene editing, and it was further demonstrated that NBCe1-B deletion in mice impairs ammoniagenesis. However, NBCe1-D only accounted for a small proportion of the expression products of SLC4A4 [29]. NBCe1-A is an important transporter chiefly expressed in S1 and S2 proximal tubule cells, and is the main mechanism of ${\mathrm{HCO}}_{3}^{-}$ efflux in the basement membrane of the renal tubule. Through in-depth research, it was found that the molar ratio of ${\mathrm{Na}}^{+}$ and ${\mathrm{HCO}}_{3}^{-}$ transported by NBCe1-A is 1:3 in the proximal renal tubules [66]. Under normal physiological conditions, including membrane potential and ion electrochemical gradient, a stoichiometric ratio of 1:3 is more conducive to NBCe1-A mediating the outward transport of ${\mathrm{Na}}^{+}$ and ${\mathrm{HCO}}_{3}^{-}$, which promotes the transport of ${\mathrm{HCO}}_{3}^{-}$ from renal tubular epithelial cells through the basal membrane to the interstitial space and, finally, its recovery into the blood.

### 4.2. Activity of NBCe1 in the Central Nervous System (CNS)

There is a complex relationship between the microenvironment of the nervous system and the activity of neurons. pH and ${\mathrm{HCO}}_{3}^{-}$ play an extremely important role in the normal function of the CNS [67]. On the one hand, the pH of the microenvironment inside and outside neurons can affect the activation of ion channels (many ion channels are pH-sensitive) and the releasement of neurotransmitters, thus affecting the excitability of the neurons. Additionally, the activity of neuronal cells (including the release of neurotransmitters and the activity of neurotransmitter receptors (such as GABA receptors and glycine receptors)) also affects the pH of the intracellular and extracellular microenvironment. Two families of acid–base transporters are expressed in the CNS [68] (NBC and NHE) which are primarily responsible for maintaining ionic and pH homeostasis. The widely expressed NHE1, NDCBE, and NBCe1 are thought to be the primary factors promoting the acid–base balance of neurons and glia in the CNS.

NBCe1 was the first NBC that to be found in the CNS (unsurprisingly, since it was also the first to be cloned among the members of the SLC4 family), and used to produce molecular samples including PCR primers, cRNA samples, and antibodies. Through the use of in situ hybridization technology [69], it can be seen that the mRNA of NBCe1 is abundant throughout the whole CNS, especially in the hippocampus, olfactory bulb, cerebellum, striatum, superior colliculus, etc. Majumdar et al. [38] studied the specific expression distribution of different variants of NBCe1 in the CNS of rats; they used Ct-specific antibodies against NBCe1-A, B, D, and E for immunofluorescence staining, and the signals were mainly distributed in the neurons, while when using Ct-specific antibodies against NBCe1-C for immunofluorescence experiments, the signals were mainly distributed in the glia. Furthermore, combining the results obtained using spliceosome-specific probes for in situ hybridization, Majumdar et al. [38] concluded that in the brains of rats, NBCe1-B and NBCe1-C are the main variants expressed. NBCe1-B is mainly distributed in the neurons, while NBCe1-C is expressed in the glial. Therefore, the expression of different spliceosomes of NBCe1 in the central system is cell-specific.

Using human NBCe1 primers, without distinguishing splice variants of NBCe1, Damkier et al. [51] identified NBCe1 mRNA in the human brain.

There is strong evidence that disturbed NBCe1 function might have pathogenic impacts. Giffard et al. [70] examined whether NBCe1 was related to the vulnerability to acid injury during ischemia, and found that DIDS-blocking transporters can inhibit acid injury in primary cultures of astrocytes. In the middle cerebral artery occlusion (MCAO) model of gerbils, the expression of NBCe1 mRNA and proteins increased in the penumbra, and was closely related to delayed cell death in the hippocampal CA1 region [71].

Acute and chronic hypoxia, along with ischemia [72], cause changes to the intracellular pH in neurons and glia (which, in turn, leads to more severe acidosis). The metabolic changes occurring during ischemia [73] (including interruption of oxygen ($O_{2}$) and glucose supply, as well as ATP synthesis disorders) can eventually lead to intracellular acidosis. At this time, the injury area maintains intracellular and extracellular the acid–base dynamic balance by regulating the transmembrane transport of $H^{+}$ and ${\mathrm{HCO}}_{3}^{-}$. During ischemia, the intracellular and extracellular microenvironments are acidic. NBCs use the electric potential difference of intracellular and extracellular ${\mathrm{Na}}^{+}$ to achieve the transmembrane transport of ${\mathrm{Na}}^{+}/{\mathrm{HCO}}_{3}^{-}$, and trans-duce ${\mathrm{Na}}^{+}$ into the cell while regulating the pH. This “pH-regulated ${\mathrm{Na}}^{+}$ influx” may be of significance in CNS ischemia/reperfusion (I/R) injury. During reperfusion [74], extracellular pH(pHo) rapidly returns to normal, while the low intracellular pH (pHi) will remain for some time (pHi < pHo). The pH gradient helps to extrude $H^{+}$ through NHEs and provides a pathway for the entry of ${\mathrm{HCO}}_{3}^{-}$ via NBCs. After ischemia/reperfusion, the gradient difference between intracellular and extracellular pH stimulates the activity of NBCs and NHEs, and activates “pH-regulated ${\mathrm{Na}}^{+}$ influx” to introduce ${\mathrm{Na}}^{+}$ into neurons. At the same time, ATP generation is impeded, and the activity of ${\mathrm{Na}}^{+}$–$K^{+}$–ATPase decreases, which further enhances the retention of the neuron ${\left[{\mathrm{Na}}^{+}\right]}_{i}$. The sustained increase in ${\left[{\mathrm{Na}}^{+}\right]}_{i}$ induces cell depolarization and triggers the abnormal release of excitatory amino acid transmitters. The increase in ${\left[{\mathrm{Na}}^{+}\right]}_{i}$ also reverses the transport mode of the ${\mathrm{Na}}^{+}/{\mathrm{Ca}}^{2+}$ exchanger (NCX), leading to ${\mathrm{Ca}}^{2+}$ influx and ${\left[{\mathrm{Ca}}^{2+}\right]}_{i}$ overload, which initiates a series of calcium-activated cell damage processes [75,76].

## 5. NBCe1-Related Diseases

NBCe1 dysfunction can cause severe ${\mathrm{HCO}}_{3}^{-}$ reabsorption problems. There are currently twelve known NBCe1 mutations [77], including eight missense mutations (R298S, S427L, T485S, G486R, R510H, L522P, A799V, and R881C), two nonsense mutations (Q29X and W516X), and two frameshift mutations (2311A and 65bp), all of which have been correlated with a variety of severe human pathological processes.

Missense and nonsense mutations in human NBCe1 were found to cause severe pRTA [78]. These patients had severely low levels of ${\mathrm{HCO}}_{3}^{-}$ in their blood, resulting in a more acidic blood pH. Sodium bicarbonate absorption defects may or may not be accompanied by proximal tubule transport defects. The absorption of bicarbonate in the proximal tubules is a combined process of apical NHE3 coupled with basolateral NBCe1-A transport. Membrane-bound carbonic anhydrase IV(CAIV) and cytoplasmic carbonic anhydrase II (CAII) catalyze the hydration and dehydration in the lumen and the cytoplasm, respectively, thereby enhancing the rate of bicarbonate absorption. The absence of additional proximal tubule transport anomalies, the unique extrarenal manifestations of development and intellectual disability; basal ganglia calcification;, ocular defects such as cataracts, band keratopathy, and glaucoma; and enamel flaws associated with incomplete amelogenesis are the only known causes of pRTA [79], and this ailment is only observed in patients who have NBCe1 mutations. Remarkably, some such patients also experience epilepsy symptoms. The phenotype of congenital NBCe1 deletion or truncation in mice is a more severe phenotype than that in humans, leading to marked volume deletion, decreased survival, hypovolemia, and colonic obstruction [80,81].

Additionally, the eye is a target organ in individuals with the NBCe1 mutations. During blinking, the normal opening of the eyelids causes intermittent loss of ${\mathrm{CO}}_{2}$, leading to acute alkalization of the anterior corneal tear coat. The increased corneal pH is usually reduced by modulating the transport of endothelial cell NBCe1-B [78]. The deletion of NBCe1-B is thought to lead to an aberrant rise in corneal pH, resulting in calcium phosphate precipitation in the central cornea. Furthermore, the inactivation of NBCe1 directly affects the homeostasis of the cornea, trabecular meshwork, and lens, leading to keratopathy, glaucoma, and cataracts. While a prior study has reported the presence of electrogenic ${\mathrm{Na}}^{+}$-based transport in the lens of toads, it was also found that NBCe1-B in the lens epithelium mediates the associated discovery in cataract patients, indicating that lens transparency is pH-dependent. However, the mechanism by which NBCe1 inactivation leads to glaucoma is currently unknown. Thus far, NBCe1 has not been detected in the retina, where NBC activity has also been reported [82]. It is possible that other transport molecules, such as NBCe2, may be responsible for the cotransport activity of the retina [83].Migraine linked to NBCe1 mutations is a primary headache, most likely produced by dysregulation of the brain’s local pH control, as opposed to a secondary headache caused by systemic homeostasis problems. Migraine is caused by specific mutation [84], such as homozygous R510H, L522P, R881C, Δ2311 A, and Δ65bp deletion mutations and heterozygous 65bp C-terminal deletion and L522P mutations in patients. In astrocytes, a moderate reduction in NBCe1-B activity may cause migraine in some heterozygotes, but not in homozygotes. Migraine in homozygotes may be due to abnormal NMD-mediated neuronal hyperactivity caused by misfolded NBCe1-B retained in the endoplasmic reticulum of astrocytes, while in heterozygotes, the wild-type mutant NBCe1-B heterooligomer retained in the endoplasmic reticulum may be involved [48].

Among the phenotypes instigated by NBCe1 mutations, NBCe1 variants show different tissue distribution and mediate different physiological functions. As previously stated, NBCe1-A inactivation is the primary cause of pRTA, while NBCe1-B inactivation is the primary cause of the ocular and neurological manifestations of the disease. Since NBCe1-C was found in rat astrocytes, the inactivation of NBCe1-C may be also implicated in the development of migraines.

## 6. NBC Inhibitors

NBCs are associated with abnormalities in fluid regulation, CNS activity, and intellectual disability. As a result, the need for special drugs that interfere with the function of NBCs is becoming increasingly urgent. Thus far, several research groups have studied pharmacological compounds for the experimental identification of the transport function of specific ${\mathrm{Na}}^{+}$ and ${\mathrm{HCO}}_{3}^{-}$cotransporters, with the potential for therapeutic application. The following is an overview of the reported NBC inhibitors, and Figure 5 shows their chemical formulae.

Stilbene sulfonic acid compounds, such as DIDS and SITS, are currently the most widely studied NBC inhibitors [85]. It is widely known that DIDS is a classical anion exchange inhibitor and that it is activated from the stilbene di-sulfonate backbone [86]. Interestingly, the isothiocyanate group of DIDS may play a major inhibitory role compared to its stilbene di-sulfonate backbone. Thus, it is believed that isothiocyanate groups interact with cell membranes to help prevent cell penetration, without the connection between stilbene di-sulfonate and anion channels. According to reports, the Ki values of DIDS and SITS range from high nanomolar to tens of micromolar concentrations, depending on cell type and ion composition; for instance, the half-maximal inhibitory concentration of DIDS for AE4 is 5 µM in *Xenopus* oocytes [87], while 500 µM inhibited NBCn1 by 95% in *Xenopus* oocytes [50], and 200 µM inhibited NBCe1 and NBCe2 by 80% in *Xenopus* oocytes [43]. A later study on erythrocyte AE1 has shown that DIDS inhibition comprises two phases: the first is a fast ionic interaction that can be inverted by the slower covalent reaction of removing DIDS and albumin; in the second interaction, DIDS blocks the transporter in all cases [88]. In the case of NBCe1, DIDS reversibly prevents NBCe1 from moving away from the outer surface of the cell by recognizing the KKMIK motif at the extracellular terminus of the potential NBCe1–TM5 complex [89]. Changes in the transporter configuration may occur when the membrane voltage lowers, which might explain why the apparent affinity of the interaction reduces as the voltage increases. Additionally, DIDS also appears to stop NBCe1 from functioning at an unknown location within the cell. Liu et al. [90] confirmed that the half-maximal inhibitory concentration of NBCe1 for DIDS is 35–40 µM, while the maximal inhibition is approximately 88%, which is more substantial than those of the ${\mathrm{Cl}}^{-}/{\mathrm{HCO}\;}_{3}^{-}$ exchangers AE1–3. However, its usefulness in vivo is limited by inhibitory actions on several other cellular components, such as AEs, ${\mathrm{Ca}}^{2+}$-activated ${\mathrm{Cl}}^{-}$-channels, carbonic anhydrases (CA), KATP-channels, and the ${\mathrm{Ca}}^{2+}$-ATPas.

The lack of selectivity of compounds such as DIDS and SITS for specific anion transport processes gives rise to more specific and selective anion transport inhibitors. One specific inhibitor of NBC, known as S0859, is an N-cyanosulfonamide compound, the availability of which offers significant new approaches to explain how NBCs are involved in different kinds of pathologies. S0859 has been revealed to block NBCs in mammalian cells; for instance, the Ki for S0859 was 1.7 µM when the transporter was developed in cultured rat ventricular myocytes [91]. S0859 inhibited the transport activity of NBCe1 with an IC50 of 9 µM in *Xenopus* oocytes [92]. The activity of NBCn1 in the MCF-7 human breast cancer cell line was also inhibited by S0859 [93]. Interestingly, while Bachmann et al. [94] suggested that S0859 was a specific NBCe1 inhibitor, it was later shown to fully inhibit all active NBCs in cardiac myocytes, without affecting AEs, NHEs, or ${\mathrm{Cl}}^{-}/{\mathrm{OH}}^{-}$-exchange activity. As such, S0859 should now provide a powerful new tool to describe the relative contribution of NHE1 and NBCs to myocardial acid extrusion in various physiological and pathological environments, especially when used with NHE1 selective inhibitors. Based on in vitro tests, it was revealed that S0859 is an effective, high-affinity, generic NBC inhibitor, and that it offers a certain improvement over the stilbene derivatives. It is worth noting that more recent studies also show that the activity of some isoforms of monocarboxylate transporter (MCT) which were heterologously expressed in *Xenopus* oocytes to transport lactate, pyruvate, and ketone bodies was reversibly inhibited by S0859, with an IC50 of 4–10 μM [92]. Therefore, S0859 seems to be an anion transport inhibitor with a wider spectral range than previously known.

In addition to DIDS and S0859, other blockers have inhibitory effects on NBCs (particularly NBCe1). The non-steroidal anti-inflammatory drug tenidap (Ki = 13–26 µM) and niflumic acid (88% inhibition by 100 µM) both inhibit the activity of NBCe1 [95]. However, in this concentration range, there are some other effects, including inhibition of both ${\mathrm{Cl}}^{-}$- and $K^{+}$-channels, as well as other transporters. Other anion transport blockers (including amiloride, oxonol dyes, and their analogs) also inhibit NBCe1 [96]. It is worth considering that the inhibition of NBCe1’s pores might include a large inner vestibule as a result of a broad-spectrum anion-blocker.

## 7. Conclusions and Future Prospects

The physiological and pathological importance of NBCe1 has been fully proven by studies of human genetics and gene knockout in mice over the past few years. This study focuses on the structural and functional characteristics of NBCe1, and also discusses the roles of NBCe1 in the kidneys and CNS, along with related dysfunctions that causes severe ${\mathrm{HCO}}_{3}^{-}$ reabsorption problems associated with a variety of severe pathophysiological conditions in humans. Furthermore, the study of the three-dimensional structure of the NBCe1 protein can provide significant insights into the molecular process that governs the stoichiometric ratio of ions transported by NBCe1, as well as the energy coupling mechanism of co-transportation of ${\mathrm{Na}\;}^{+}$and ${\mathrm{HCO}}_{3}^{-}$. In addition, researchers should explore and develop targeted drugs for NBCe1 and its related pathways for interventions to treat genetic diseases.

## Figures and Tables

**Figure 1 brainsci-11-01276-f001:**
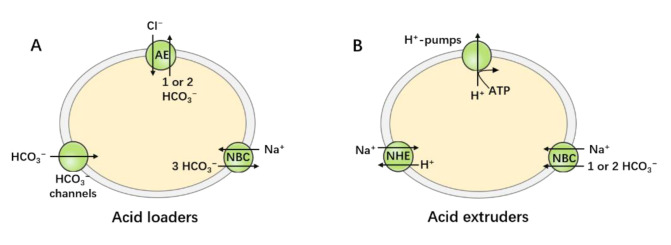
Acid–base transporters. Acid–base homeostasis is achieved by the balance of the action of the acid-loading and acid-extruding mechanisms. (**A**) The acid loaders include AEs, electrogenic NBCs, and ${\mathrm{CHO}}_{3}^{-}$ channels that permit the passive influx of $H^{+}$ or efflux of ${\mathrm{CHO}}_{3}^{-}$. (**B**) The acid extruders include $H^{+}$-pumps, NHEs, and both electrogenic and electroneutral NBCs.

**Figure 2 brainsci-11-01276-f002:**
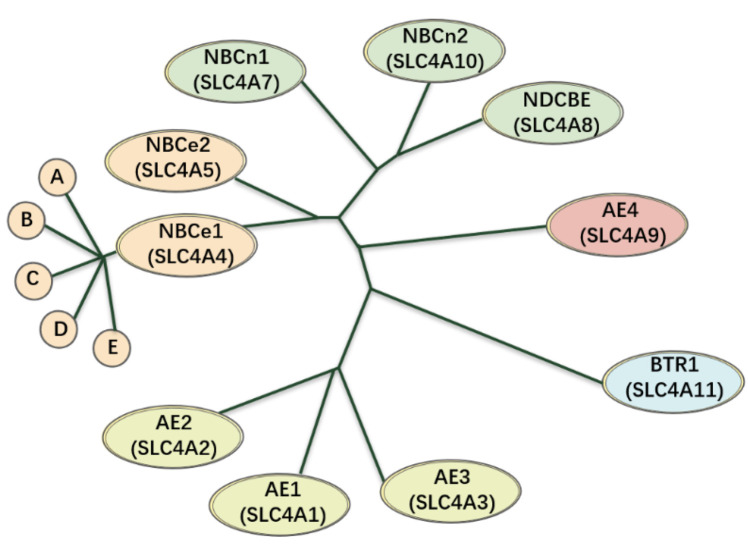
An evolutionary diagram using the SLC4A family as a phenotype. The anion exchangers (AE1–3) of this family seems to be almost branched from the same ancestor, being ~53–56% indistinguishable from one another at the amino acid level. Meanwhile, NBCe1 and NBCe2 are derived from a form that was intermediate between AEs and electroneutral NBCs, which are ~53% identical to one another, ~39–50% identical to the electroneutral NBCs, and ~28–34% identical to the AEs. In addition, NBCn1, NBCn2, and NDCBE are ~71–76% identical to one another and ~30–34% identical to the AEs. AE4 and BTR1 are not closely related to any other species in the phylogeny, including one another.

**Figure 3 brainsci-11-01276-f003:**
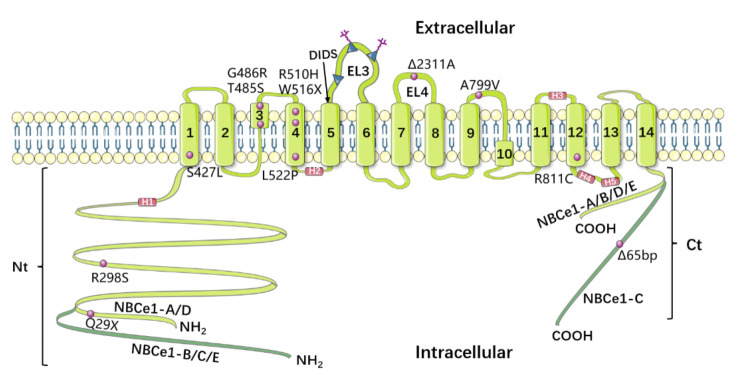
Model of NBCe1′s topology, based on Huynh et al. [28]. NBCe1-A and -D have an exceptionally short Nt, whereas NBCe1-C has an exceptionally long Ct. The “KKMIK” motif is located at the extracellular end of TM5, and serves as a binding site for DIDS. The third extracellular loop has three possible N-glycosylation sites (EL3). The fourth extracellular loop (EL4) is crucial for NBCe1’s electrogenicity. The purple circles represent 12 identified NBCe1 mutations, while the purple squares show 4 amphipathic helices (H1–4) and a short cytoplasmic helix (H5).

**Figure 4 brainsci-11-01276-f004:**
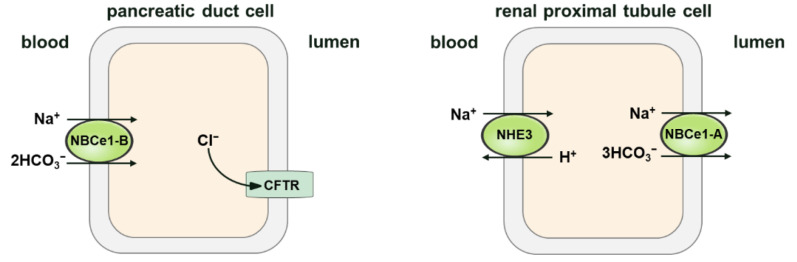
Schematic diagram illustrating the modes of NBCe1-A/B in the pancreatic duct and renal proximal tubule cells. The ${\mathrm{Na}}^{+}/{\mathrm{HCO}}_{3}^{-}$ cotransporter (NBC1-B) operates in influx mode in the pancreas, and has a probable stoichiometry of 2 ${\mathrm{HCO}}_{3}^{-}$ per ${\mathrm{Na}}^{+}$. In renal proximal tubule cells, the ${\mathrm{Na}}^{+}/{\mathrm{HCO}}_{3}^{-}$ cotransporter (NBC1-A) performs in efflux mode, and has a stoichiometry of 3 ${\mathrm{HCO}}_{3}^{-}$ per ${\mathrm{Na}}^{+}$. The primary structural difference between NBCe1-A and NBCe1-B is in the amino-terminus (residues 1–41 of NBCe1-A are different from residues 1–85 of NBCe1-B). The difference in stoichiometric ratio between endogenous NBCe1-A and NBCe1-B may be the result of cell specificity, but the difference in the N-terminal structure is not. For example, When Slc4a4-deficient renal tubular collecting duct epithelial cells are heterologously expressed in NBCe1-A and NBCe1-B, the stoichiometric ratio of both is 2; while the stoichiometric ratio of both is 3 in Slc4a4-deficient proximal tubular epithelial cells, which indicates that the stoichiometric ratio of NBCe1 is cell-dependent.

**Figure 5 brainsci-11-01276-f005:**
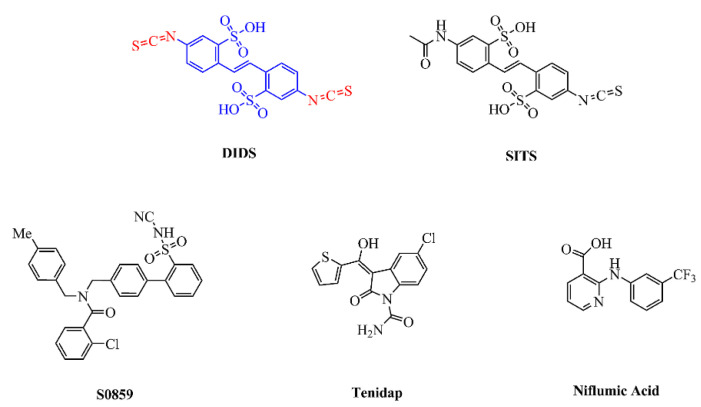
Chemical structures of published NBC inhibitors, including stilbene sulfonic acid compounds (DIDS, SITS), S0859, the non-steroidal anti-inflammatory drug tenidap, and niflumic acid. The structure of DIDS includes a stilbene di-sulfonate backbone (blue) and two isothiocyanate groups (red).

## Data Availability

No new data were created or analyzed in this study. Data sharing is not applicable to this article.

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
