# Peer review of "Understanding the Functional Expression of Na+-Coupled SLC4 Transporters in the Renal and Nervous Systems: A Review"

_brainsci, 2021, doi:10.3390/brainsci11101276_

Round 1

Reviewer 1 Report

Du et al. wrote a very interesting article describing “??+/???− cotransporter (NBCe1) functional expression in the renal and nervous system”. The manuscript represents an interesting way to discover new scenarios for ion involvement in renal and nervous functions. I suggest only several minor revisions needed to update and improve the reliability of the paper:

  • The “Introduction” section lacks of sufficient and updated references. I suggest to add the importance of ion channel and transporters in other areas of nervous system, such as the retina. Regarding this, I suggest to cite the recent publication PMID: 33374679.
  • The manuscript should be better formatted. For example, the authors should insert at least a space between the end of figure captions and the text.
  • Finally, manuscript requires English revisions and typos correction.

Author Response

Point 1: The “Introduction” section lacks of sufficient and updated references. I suggest to add the importance of ion channel and transporters in other areas of nervous system, such as the retina. Regarding this, I suggest to cite the recent publication PMID: 33374679.

Response 1: Thank you for the suggestion we have made changes to the references [8, 9,10,11,12,13]. In addition, we have emphasized the significance of ion channels and transporters in the retina. (Section 1-Introduction, line 55-74 please, Section 5-NBCe1-related diseases, line 386-409) and also referred to the suggested publication (please See Reference #13).

Point 2: The manuscript should be better formatted. For example, the authors should insert at least a space between the end of figure captions and the text.

Response 2: We greatly appreciate the suggestion. Corrections are made in the manuscript.

Point 3: Finally, manuscript requires English revisions and typos correction.

Response 3: We very much appreciate the careful reading of our manuscript and valuable suggestions of the reviewer. We have carefully considered the comments and have revised the manuscript accordingly.  The MS has been carefully checked by an English spoken colleague, and also undergone English language editing by MDPI. The text has been checked for correct use of grammar and common technical terms, and edited to a level suitable for reporting research in a scholarly journal.

Reviewer 2 Report

The review is not just on NBCe1 but rather Na-cuopled NBC's so the title should change

Regarding the structure of NBCe1, the CryoEM structure of NBCe1 has been determined (Nature Communications volume 9, Article number: 900 (2018)) . This paper should be reviewed in detail.

Author Response

We very much appreciate the careful reading of our manuscript and valuable suggestions of the reviewer. We have carefully considered the comments and have revised the manuscript accordingly. 

The comments can be summarized as follows:

Point 1: The review is not just on NBCe1 but rather Na-cuopled NBC's so the title should change.

Response 1: We appreciate this concern immensely and have changed the title of this article as you suggested “Understanding the functional expression of Na+-Coupled SLC4 Transporters in renal and nervous system: A review”.

Point 2: Regarding the structure of NBCe1, the CryoEM structure of NBCe1 has been determined (Nature Communications volume 9, Article number: 900 (2018)). This paper should be reviewed in detail.

Response 2: We thank the reviewer for the concerns. We have modified the structure of NBCe1 (See Section 3.1-NBCe1 (SLC4A4), Figure 3) and cited the suggested article in figure-3 ligand [28]. We have added the details and reviewed the above article carefully (See Section 3.1-NBCe1 (SLC4A4), line 127-135, and Section 5-NBCe1-related diseases, line 386-409).

Round 2

Reviewer 2 Report

The requested changes have been made.

Author Response

Thanks for your valuable comment.